# Effect of Weight Self-Stigma on Quality of Life and Dietary Habits among Adult Students in Riyadh, Saudi Arabia

**DOI:** 10.3390/healthcare11121754

**Published:** 2023-06-15

**Authors:** Alanoud Aladel, Badeeah Dakhakhni, Yara Almuhtadi, Azzah Alsheweir, Sadeem Aljammaz

**Affiliations:** Community Health Sciences, College of Applied Medical Sciences, King Saud University, Riyadh 11451, Saudi Arabia; 442202861@student.ksu.edu.sa (B.D.); yalmuhtadi@ksu.edu.sa (Y.A.); aalshuweir@ksu.edu.sa (A.A.); saljammaz@ksu.edu.sa (S.A.)

**Keywords:** weight self-stigma, WSS, quality of life, QOL, dietary habits, adult students, BMI, Saudi Arabia

## Abstract

Weight self-stigma (WSS) is a personal experience of negative self-evaluations, perceived discrimination, and shame about body weight. Studies suggested that WSS could negatively affect quality of life, eating behavior, and psychological outcomes. WSS has been linked with a number of obesogenic health outcomes that complicate weight loss interventions. Thus, this study aimed to examine the effect of WSS on the quality of life and dietary habits among adult students. A sample of 385 students from Riyadh universities participated in this cross-sectional study and completed three online questionnaires: the WSS questionnaire, the WHO quality of life questionnaire, and a dietary habit questionnaire. The average age of participants was 24 ± 6.74, and the majority were female (78.4%). Results demonstrated a negative association between all QOL domains and WSS (*p* < 0.001). Moreover, higher BMI is associated with increased self-devaluation and fear of enacted stigma (*p* < 0.001). There was also a negative link between both quality and quantity of food with WSS (*p* < 0.01). No significant difference was seen in study outcomes concerning gender. The findings of this study suggest the importance of increasing awareness about the negative impact of WSS and developing social policies to prevent or decrease it. Additionally, multidisciplinary teams, especially dietitians, should be more aware of WSS when dealing with overweight and obese individuals.

## 1. Introduction

The term “stigma” refers to a mark, a condition, or a position that is prone to devaluation [1,2]. In the early literature, stigma has a substantial scope of interest in sociology, particularly with mental illness and HIV/AIDS conditions [3]. However, in the last 14 years, the traditional domain of stigma went beyond mental illness and expanded to reach various health topics and conditions, including obesity and body weight [4,5,6]. There are two types of stigmas related to weight: public stigma and self-stigma. The term “public stigma”, or “external stigma”, refers to the perception of discrimination by others in social contexts, such as relationships or the workplace. The definition of self-stigma, also known as internalized stigma, is the acceptance of stereotypes associated with public stigma and the application of those stereotypes to oneself, which in turn fuels unfavorable feelings and ideas about being overweight or obese. The most important type of stigma is weight self-stigma (WSS). It is a personal experience of negative self-evaluations, a sense of shame, and perceived weight discrimination, particularly in those who are overweight or obese [7,8]. Society’s perception of weight stigma continues to exist because it is believed to be linked to personal self-control and responsibility, rather than to a combination of psychological, physiological, and genetic factors that play a role in developing obesity [9,10]. This negative belief reinforces obese individuals as being lazy, unmotivated, lacking in discipline, and more likely to suffer from social rejection and unfair treatment [11,12,13,14]. This defies the principles of social equality and diversity. Therefore, fighting weight stigma is crucial for human rights and for tackling obesity [15]. The prevalence rate of WSS in Saudi Arabia (SA) was 46.4%, and obesity was the strongest factor associated with WSS [16]. Many aspects of life can be negatively affected by WSS and cause negative psychological, behavioral, and social outcomes. First of all, several studies evaluated the relationship between WSS and psychological distress in overweight and/or obese people. A cross-sectional study conducted in SA showed that stigmatization of the weight of oneself was significantly associated with depression, anxiety, and stress [17]. Another study showed that lower WSS total scores were a predictor of lower psychological distress [18]. Secondly, poorer quality of life (QOL) was linked with internalized weight stigma in adults who participate in weight loss-seeking programs [19]. According to the WHO QOL group, QOL was explained as “an individual’s perception of their position in life in the context of the culture and value systems in which they live, and in relation to their goals, expectations, standards, and concerns” [20]. A study found that lower WSS scores were linked with better QOL [18]. Another study conducted in SA found that participants who experience high WSS levels have lower QOL than participants with lower WSS levels [21]. Thirdly, an interesting finding is that people’s eating habits could be negatively impacted by weight stigma, which could have an impact on their overall health [22]. Research showed participants who were overweight or obese and experienced weight stigma were frequently linked with unpleasant self-reported diet quality [22]. According to Major et al. (2014), overweight participants who experienced weight stigma consumed more calories, which was around 4% more calories per day than what is suggested for an average adult. Additionally, those who faced weight stigma reported eating more fried and fast food [23]. Fourthly, the literature emphasizes how WSS has a negative impact on physical activity (PA) levels. PA avoidance and WSS have been linked in research repeatedly [19,21]. Finally, several studies have provided strong evidence that suffering from weight stigma had a positive correlation with BMI [24,25]. Therefore, the main aim of this study was to examine the effect of WSS on the QOL and dietary habits among Saudi adult female and male students in Riyadh region. Further, to explore whether high BMI is associated with the existence of weight self-stigma, low quality of life, and/or poor dietary habits.

## 2. Materials and Methods

### 2.1. Study Design and Participants

A cross-sectional study was undertaken between September 2022 and February 2023. Based on a confidence interval of 95%, an acceptable margin of error of 5%, and a nonresponse rate of 20%, a practical random sample of 385 adult students was selected. An online survey was delivered to eligible participants via the social media networks and apps WhatsApp (version no. 23.9. 77), Snapchat (version no. 12.39.0.38), and Twitter (version no. 9.89.0). To enable anonymous, voluntary participation, a self-report questionnaire was created using Google Forms and made available throughout the study. Adult students (*n* = 385) included male (*n* = 83, 21.6%) and female (*n* = 302, 78.4%) university students (M age = 24, SD = 6.74) from 10 different universities in Riyadh, Saudi Arabia, who have given their consent prior to enrolling in the study. The study population was Arabic-speaking male and female university students aged 18 years and above, attending universities in Riyadh region, Saudi Arabia.

### 2.2. Socio-Demographic Variables

After the enrolment consent section, participants were asked to answer demographic data questions (as the first section of the questionnaire) which included: questions about age, sex, nationality, educational level, name of the university studied at, marital status, monthly salary, place of residence, weight, and height. Body mass index (BMI) was calculated using the Quetelet equation, calculated by weight (kg) divided by height (m^2^), and recruited students were classified as underweight (BMI < 18.5 kg/m^2^), normal weight (18.5 ≤ BMI < 25.0 kg/m^2^), overweight (25.0 ≤ BMI < 30.0 kg/m^2^), obesity class I (30.0 ≤ BMI < 34.5 kg/m^2^), obesity class II (35.0 ≤ BMI < 39.5 kg/m^2^), obesity class III (BMI ≥ 40.0 kg/m^2^) conforming to WHO guidelines [26,27].

### 2.3. Weight Self-Stigma Questionnaire

The weight self-stigma questionnaire (WSSQ), a valid and reliable indicator made up of a 12-item Likert-type scale of weight-related self-stigmatization, was included in the second portion [28]. The responses ranged from “strongly disagree” to “strongly agree” on a scale of 1 to 5. It includes two subscales that assess fear of social stigma and self-devaluation related to weight. Therefore, it indicates that the Arabic WSSQ is a reliable tool for evaluating weight-related self-stigma in Arabic-speaking individuals (α = 0.982) [28].

### 2.4. The WHO Quality of Life (QOL) Questionnaire

The WHO Quality of Life (WHOQOL)-BREF, which was created to assess QOL, is translated into Arabic in the third section to measure the quality of life. The section includes twenty-six items in total, distributed as follows: seven for the physical health domain, six for the psychological health domain, three for the social ties domain, and eight for the environmental domain. Additionally, a two-item self-rating of overall QOL and general health satisfaction. Each item is rated on a scale of 1 to 5, with higher scores indicating a better quality of life. It is a self-administered questionnaire that reported significant reliability and validity indices among the Arab general population (α ≥ 0.7) [29,30].

### 2.5. Dietary Habits Questionnaire

The questionnaire’s final section is divided into three parts. Three questions about general eating habits were covered in the first section. Questions about dietary food quality and quantity were covered in the second and third sections, respectively. The seven questions about food quantity were on a scale of 0 to 3, depending on how frequently a food item was consumed per day or week, while the five questions about food quality were scaled from 1 to 5 (“strongly disagree” to “strongly agree”) [31]. This Arabic dietary habits questionnaire was adopted from Alhusseini & Alqahtani’s [31] study and was approved for its content validity. The authors translated and adjusted the original version published in earlier studies to meet the Saudi population’s dietary requirements [32,33].

### 2.6. Statistical Analysis

The statistical analysis was performed using SPSS Statistics version 25.0. Descriptive and comparative statistics were used to analyze the data. Qualitative variables are reported as frequency and percentage, while quantitative variables are reported as mean ± standard deviation (SD), unless ± standard error (SE) of the mean is specified. Two-tailed independent samples *t*-tests were used to compare mean differences between two independent groups. To evaluate differences between multiple groups, a one-way analysis of variance (ANOVA) was used. The data were checked for normality using the Shapiro–Wilks criteria and a visual inspection of QQ-plots. Bivariate Spearman correlation analyses were used to analyze associations between non-parametric variables. *p* values of <0.001, <0.01, or 0.05 were considered statistically significant.

## 3. Results

### 3.1. Baseline Characteristics of the Sample

The baseline characteristics of the sample are described in detail in Table 1. The majority of the students who volunteered to complete the questionnaire were female (302 (78.1%) of 385), with only 21.6% males recruited (*n* = 83). Most of the students were in bachelor’s and postgraduate education degrees (79.5%) with a mean age of 24 ± 6.74 years. The majority of the students were from King Saud University (KSU) (64.9%), 14.3% were from Princess Nourah Bint Abdul Rahman University (PNU), 7.5% were from Dar Al Uloom University and 13.3% were from other universities in Riyadh region. BMI calculation showed a considerable proportion of 170 (44.2%) of students with normal weight, while students who were classified as overweight comprised 23.9% (*n* = 92), and 12.2% (*n* = 47), 5.5% (*n* = 21) and 4.4% (*n* = 17) with obesity class I, class II, and class III, respectively (Table 1).

### 3.2. Prevalence of WSS

The respondents who reported feelings of inferiority had the highest prevalence of WSS, as measured by self-devaluation statements (29.4%), followed by those who expressed feelings of self-blame (24.4%), guilt (19.7%), poor self-control, and weakness (17.1%). For fear of enacted stigma statements, 16.9% believed that others thought they lacked self-control because they did not control their weight, 18.2% reported a perceived lack of others’ sympathy, 15.8% thought that others blamed them for weight problems, 12.7% felt insecure, and 7.8% reported enacted stigma from discrimination (Table 2). No significant differences were reported across sex.

### 3.3. Self-Devaluation

According to the independent samples *t*-test, the mean self-devaluation did not differ significantly between males and females, though females comprised the majority of the collected sample (78.4%). Moreover, no significant difference between self-devaluation and adult students’ educational level was shown, according to a one-way ANOVA. A one-way ANOVA showed a significant difference between self-devaluation and BMI classifications among recruited students (*p* < 0.001). Students with obesity class II had greater mean scores for self-devaluation than individuals in other weight categories (3.9 ± 0.84) (Table 3). There was a significant difference between the normal weight and all other weight groups (*p* < 0.01). The overweight and obesity groups scored more negative self-devaluation than the underweight and normal weight groups. The scatterplot (Figure 1) illustrates the linear relationship between the subject’s mean scores on self-devaluation and BMI. There is a positive significant correlation between self-devaluation and BMI [r = 0.47, *p* < 0.001] indicating higher levels of self-devaluation with being overweight and obese. A bar chart (Figure 2) shows self-devaluation mean and BMI of underweight (2.4 ± 0.9), normal weight (2.8 ± 0.89), overweight (3.5 ± 0.88), obese class I (3.8 ± 0.63), obese class II (3.9 ± 0.84), and obese class III students (3.5 ± 0.93). Compared to normal weight students, the mean scores for self-devaluation tend to be significantly higher in overweight (*p* < 0.001) and obese (class I (*p* < 0.001), class II (*p* < 0.001), class III (*p* < 0.01)) students.

### 3.4. Fear of Enacted Stigma

No significant difference in the fear of enacted stigma means between males and females was determined according to an independent samples *t*-test, though females comprised the majority of the collected sample (78.4%). A one-way ANOVA showed no significant difference between the fear of enacted stigma and the students’ educational level. Moreover, a one-way ANOVA showed a significant difference between the fear of enacted stigma and BMI classifications (*p* < 0.001). Adult students who were classified with obesity class III had higher scores in fear of enacted stigma (3.43 ± 0.90) (Table 3). There was a substantial difference between normal weight and all other weight groups (*p* < 0.001). The overweight and obesity groups scored more on negative fear of enacted stigma than the normal weight group. Moreover, all obesity groups scored more on negative fear of enacted stigma than the overweight group. The scatterplot (Figure 3) illustrates the linear and positive relationship between the adult students’ mean scores on the fear of enacted stigma and BMI [r = 0.39, *p* < 0.001] suggesting a significant and moderate increase in the fear of enacted stigma with higher BMI measures. A bar chart (Figure 4) shows the fear of enacted stigma means and BMI of underweight (2.1 ± 0.91), normal weight (2.2 ± 0.83), overweight (2.7 ± 0.93), obesity class I (2.9 ± 0.86), obesity class II (3.4 ± 0.90), and obesity class III students (3.2 ± 0.87). The fear of enacted stigma is confirmed to be significantly higher among the overweight students (*p* < 0.001), class I obese students (*p* < 0.001), class II (*p* < 0.001), and class III (*p* < 0.001) compared to the normal weight students.

### 3.5. Quality of Life and WSS

According to the independent samples *t*-test, there was no significant difference in the QOL mean scores between males and females. No significant difference between the QOL and educational level was observed. However, as shown by one-way ANOVA, there was a significant difference between QOL and BMI categories (*p* < 0.001). Individuals with normal BMI had significantly higher QOL scores compared to overweight students (*p* < 0.05), students with obesity class I (*p* < 0.05), and obesity class II (*p* < 0.001). The scatterplot (Figure 5a) demonstrates the linear and negative association between the adult students’ mean scores in QOL and BMI [r = 0.11, *p* < 0.05], indicating that QOL was significantly reduced with a higher BMI. A bar chart (Figure 5b) shows the quality of life and BMI of underweight students (16.1 ± 3.86), normal weight (18 ± 3.29), overweight (16.6 ± 3.30), obese class I (16.1 ± 3.78), obese class II (14.4 ± 3.51), and obese class III (16.5 ± 3.88). Scores in QOL tend to be significantly higher among normal weight students compared to overweight (*p* < 0.05), class I obese (*p* < 0.05), and class II obese students (*p* < 0.001). However, scores for class III obese students did not report a significant difference from normal weight students, according to the correlation between the QOL and WSS. The mean ± SD of each subscale was described in Table 4. WSS correlated negatively with physical health domains [r = 0.35, *p* < 0.001], psychological health domains [r = 0.41, *p* < 0.001], social relationships domains [r = 0.24, *p* < 0.001], environmental health domains [r = 0.23, *p* < 0.001] and overall QOL and general health domains [r = 0.40, *p* < 0.001]. The scatterplot (Figure 5c) illustrates the linear and negative association between adult students’ mean scores on the QOL and total WSS [r = 0.40, *p* < 0.001]. This means that QOL was significantly reduced in students who reported having WSS.

### 3.6. Dietary Habits and WSS

The frequency and percentage of the first three questions in the dietary habits questionnaire were shown in (Table 5). The majority of the students reported having only 1–3 meals a week from restaurants (68.8%) and 70.9% were eating home-cooked meals 4–6 times a week or on a daily basis. However, the overall habits of eating healthy foods were reported as poor by 34% of the respondents, while the rest were distributed between fair (21%) to excellent overall eating habits (4.4%).

### 3.7. Quality of Food and WSS

The quality of food mean scores did not differ significantly between males and females as shown by the independent samples *t*-test (*p* > 0.05). No significant difference between the quality of food and educational level was observed. Moreover, a one-way ANOVA showed no significant difference in the quality of food among different BMI categories. According to the relationship between the quality of food and WSS, the scatterplot (Figure 6) illustrates the linear and negative relationship between the subject’s mean scores on the quality of food and total WSS. [r = 0.16, *p* < 0.01]. This demonstrates that those who expressed WSS saw a significant reduction in food quality.

### 3.8. Quantity of Food and WSS

As shown by the independent samples *t*-test, the quantity of food mean scores did not differ significantly between males and females (*p* > 0.05). However, there was a significant difference between the quantity of food and the students’ educational level (*p* < 0.001). However, no significant difference in the quantity of food among BMI groups was shown by a one-way ANOVA. According to the relationship between the quantity of food and weight self-stigma, the scatterplot (Figure 7) illustrates the linear and negative relationship between the subject’s mean scores on the quantity of food and total WSS [r = 0.19, *p* < 0.001]. This demonstrates that those who expressed WSS saw a significant reduction in food quantities.

## 4. Discussion

WSS has been associated with unfavorable attitudes among overweight or/and obese individuals, which can negatively affect several aspects of life, including psychological, behavioral, and social aspects. This study revealed four key findings. Firstly, QOL was significantly reduced among those who reported greater scores in WSS. Secondly, higher scores in WSS were significantly associated with reduced quality of food. Thirdly, as the WSS score increased, the quantity of food decreased. Finally, higher BMI was linked with greater scores in WSS, and lower scores in QOL.

According to the WSS and QOL domains, WSS correlated negatively with physical and psychological health, environmental health, social relationships, overall quality of life, and general health domains in our study. In other words, all QOL domains were substantially reduced in those who showed having WSS. Our findings support the results of Khodari et al. (2021) which showed that WSS is linked to lower physical, psychological, social, and environmental aspects in adult students who reported WSS [21]. Moreover, Farhangi et al. (2016) found that lower total WSS scores were associated with better QOL [19]. Another study investigating WSS among 170 participants with obesity in Spain confirmed the positive association between WSS and BMI, depression, and anxiety [34]. A recent study in Hong Kong concluded a significant and adverse association between WSS and poor health-related quality of life (r  =  −0.28 to −0.61) among overweight individuals [35].Moreover, current research focused on assessing the prevalence of WSS in adults aged 18 and older (*n* = 3821) concluded that more than half of the sample (57%) had weight stigma and that the odds of WSS were the highest among overweight and obese adults [36]. Interestingly, there was a campaign about the end of weight stigma on World Obesity Day 2018 in the UK, which aimed to shed light on the ubiquity and seriousness of weight stigma [37], which indicates that this is a global problem and the importance of tackling it.

Results of the present study demonstrated the potential of WSS to adversely affect individuals’ eating habits, which could, in turn, have negative effects on their general health [38]. Our findings show that overall eating habits significantly reduced in quality for students who scored higher WSS [r = 0.17, *p* < 0.001]. Likewise, there was a significant adverse correlation between the quality of food and the total score of WSS [r = 0.16, *p* < 0.01]. This means that individuals who suffer from WSS might consume a significantly poorer quality of food. These findings are in line with a previous study demonstrating that overweight or obese participants who frequently experience weight stigma were consuming significantly lower-quality food [38]. To our knowledge, there is not enough research about the association between WSS and the quality of food or healthy eating choices. However, many studies have shown that WSS affected eating behaviors [8,38]. According to Vartanian et al. (2016), WSS is linked to binge eating, skipping meals, less motivation to diet, uncontrolled eating, and poor eating behaviors [38].

The current literature highlights the correlation between WSS and the quantity of food. Our results demonstrate a significant negative link between the quantity of food and the total score of WSS [r = 0.20, *p* < 0.001]. This particular finding could be a result of people’s misinformed efforts, particularly those who are trying to lose weight by consuming fewer quantities regardless of the quality or calorie content of the food. Another possible explanation of our results of lower quantity of food consumption among individuals with weight stigma would be that these individuals were motivated to represent themselves positively in front of people, to avoid any potential negative judgments from them. Similarly, Meadows et al. (2019) showed that people who had greater levels of WSS consumed fewer calories from snacks after being exposed to a weight-stigma prime [39]. On the other hand, Major et al. (2014) reported that exposure to weight stigma caused overweight individuals to consume 4% more calories than the daily caloric intake suggested for an average adult, regardless of the quantities [23]. Moreover, people with weight-related stigma reported eating more fried and fast food [40].

Finally, a one-way ANOVA showed a significant relationship between BMI classifications and total WSS scores in our present study (*p* < 0.001). We found that BMI is positively correlated with self-devaluation [r = 0.47, *p* < 0.001] and the fear of enacted stigma [r = 0.39, *p* < 0.001]. A previous study provided evidence that experiencing weight stigma positively correlated with higher BMI [r = 0.06, *p* < 0.05] [41]. However, another study found no association between WSS and overweight individuals compared to obese individuals [21]. The higher rates of WSS in overweight and obese individuals might be related to social attributions about the reasons for obesity, which, in turn, could affect weight stigma expressions substantially. Puhl and Heuer (2010) reported that obese adults and children in the United States experience WSS because their higher weight is perceived to be caused by behaviors, such as lack of exercise and overeating, that are under their personal control [42]. According to the World Health Organization, the overall prevalence of overweightness in SA is estimated to be 68.2%, and obesity is 33.7% [43]. Therefore, as supported by this paper, WSS is expected to be a widespread phenomenon and is associated with a significant correlation with a high BMI.

In the present study, normal weight BMI comprises 44.2% of the sample, the overweight group comprises 23.9%, and the obese group comprises 22%. In Saudi Arabia, the prevalence of obesity in 2020 in males and females aged between 18 and 19 years old was 14.1%, and 14.8% for 20 to 29 years old, which are the lowest percentages among all other age groups, in which the highest rate was among the 50 to 59 years age group (32.8%) [43]. Furthermore, the prevalence of overweightness in females aged 30 to 59 years old was 31.7%, and 28% for the age group of 18 to 29 years. In addition, the prevalence of obesity in females between 30 and 59 years old was 55.5%, compared to only 21.5% among females between 18 and 29 years [43]. Moreover, a study by Alqahtani et, al. (2022) has shown that normal BMI was the highest among Saudi females aged 18–29 years old (45%) and females with obesity comprised the lowest percentage of this age group (21.5%) [44]. These findings could explain why, in this study, the normal weight group comprised the highest distribution of 44%, in contrast to only 22% of the group with obesity, seeing that females comprised the majority of the study sample (78.4%).

The present study has certain limitations. Firstly, the study’s cross-sectional nature makes it difficult to better clarify the causal association between WSS and other factors and/or predictors. Secondly, the limited sample size may limit the ability to generalize the results to Saudi students and to other significant regions in Saudi Arabia. Considering that the survey was anonymously distributed to the target population, most respondents were females (78.4%), making it difficult to generalize our results, particularly to males, which requires further investigation. Addressing the application of self-reported questionnaires, this study was expressed as one-stage research, since no clinical assessments or interviews were applied. Moreover, no further assessment was established for some variables that could have a significant impact (such as psychological distress, success during studies, employment, way of living, etc.). Despite the fact that self-reported questionnaires are feasible and affordable tools, they might increase the probability of overestimation and bias. We suggest that future studies look for the effects of WSS on nutritional status, lifestyle change, physical activity level, and the effectiveness of weight loss interventions. In addition, we advocate for more research about the impact of cognitive behavioral therapy on WSS.

## 5. Conclusions

In conclusion, the current paper presents the negative impacts of WSS on QOL and dietary habits. It is important to increase awareness about the adverse effects of WSS on many aspects of life, and not all people respond to this as a motivator to lose weight. We recommend developing social policies aiming to prevent weight stigma, especially in schools and universities. Further, we view that multidisciplinary teams—particularly dietitians—should be more aware of WSS when they deal with overweight and obese people. Therefore, researchers must identify interventional programs and strategies that include the reduction of WSS as a key therapeutic objective of obesity treatment programs. Combined interventions that aim to promote weight loss and healthy behavioral change with cognitive behavioral therapy and mindfulness could be beneficial.

## Figures and Tables

**Figure 1 healthcare-11-01754-f001:**
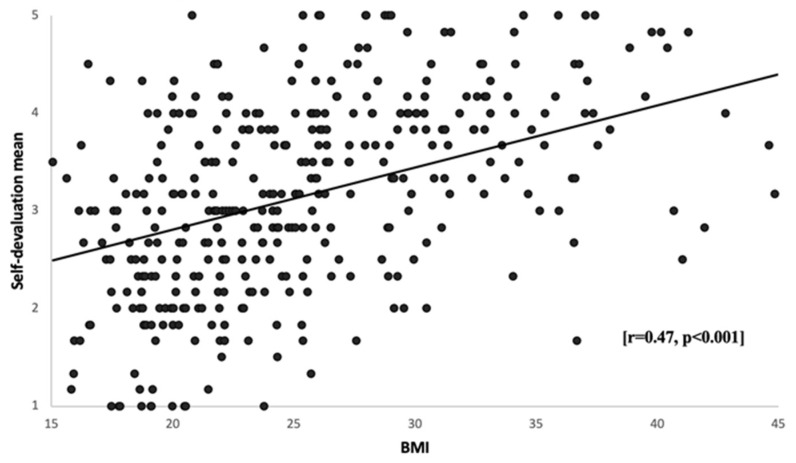
Association between BMI and self-devaluation mean: Scatter plots showing a correlation between students’ mean scores on self-devaluation and BMI, *n* = 382.

**Figure 2 healthcare-11-01754-f002:**
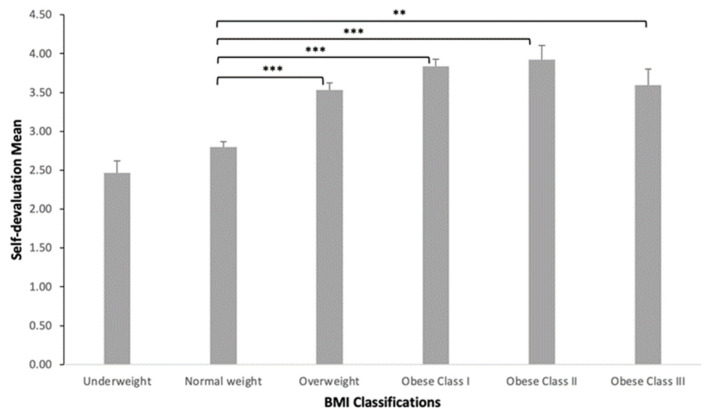
The relationship between BMI classifications and self-devaluation mean: Bar chart showing self-devaluation mean and BMI of underweight (2.4 ± 0.9), normal weight (2.8 ± 0.89), overweight (3.5 ± 0.88), obese class I (3.8 ± 0.63), obese class II (3.9 ± 0.84), and obese class III students (3.5 ± 0.93). Data are presented as mean ± standard error of the mean. A one-way ANOVA was used to determine the mean differences between self-devaluation and BMI. ** *p* < 0.01, *** *p* < 0.001.

**Figure 3 healthcare-11-01754-f003:**
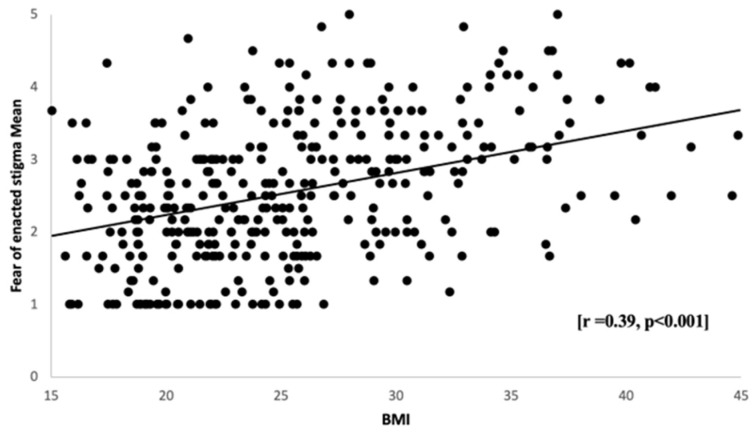
Relationship between BMI and the fear of enacted stigma mean scores: Scatter plots showing a correlation between students’ mean scores on fear of enacted stigma and BMI. *n* = 382.

**Figure 4 healthcare-11-01754-f004:**
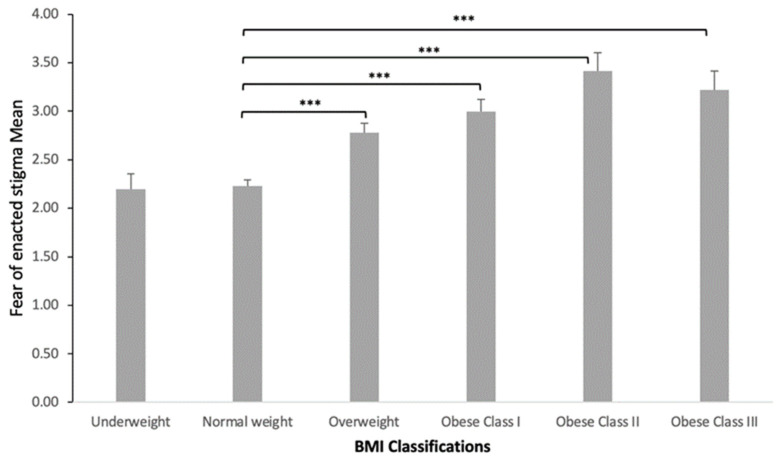
Mean scores in the fear of enacted stigma for each BMI category: Bar chart showing mean scores in the fear of enacted stigma for underweight students (2.1 ± 0.91), normal weight (2.2 ± 0.83), overweight (2.7 ± 0.93), obese class I (2.9 ± 0.86), obese class II (3.4 ± 0.90), and obese class III (3.2 ± 0.87) students. Data are presented as mean ± standard error of the mean. Statistical mean differences between mean scores in the fear of enacted stigma and BMI categories were determined via a one-way ANOVA. *** *p* < 0.001.

**Figure 5 healthcare-11-01754-f005:**
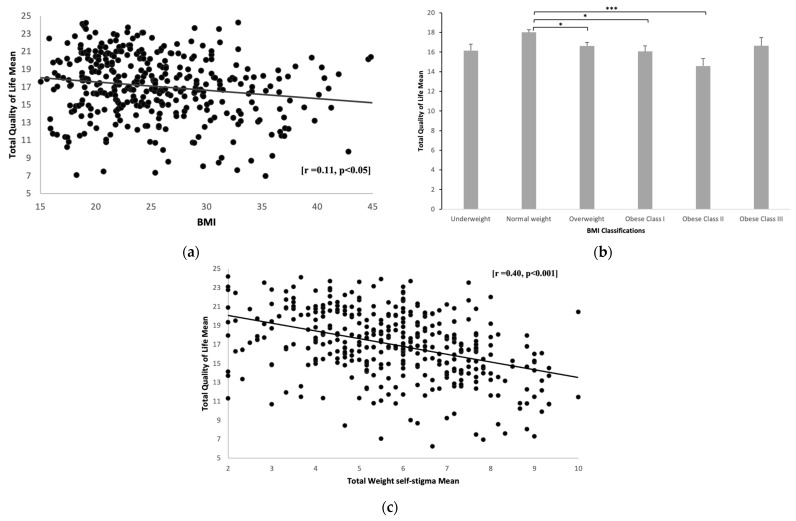
Quality of life with BMI and weight self-stigma: (**a**) scatter plots showing a correlation between adult students’ total mean scores in QOL and BMI, *n* = 382. (**b**) Bar chart showing the mean differences between total mean scores in QOL for different BMI groups, underweight students (16.1 + 3.86), normal weight (18 + 3.29), overweight (16.6 + 3.30), obese class I (16.1 + 3.78), obese class II (14.4 + 3.51), and obese class III (16.5 + 3.88). Data are presented as mean ± standard error of the mean. A one-way ANOVA was used to determine the mean differences of QOL mean scores among BMI groups. * *p* < 0.05, *** *p* < 0.001. (**c**) Scatter plots showing an association between adult students’ total mean scores in QOL and WSS. *n* = 385.

**Figure 6 healthcare-11-01754-f006:**
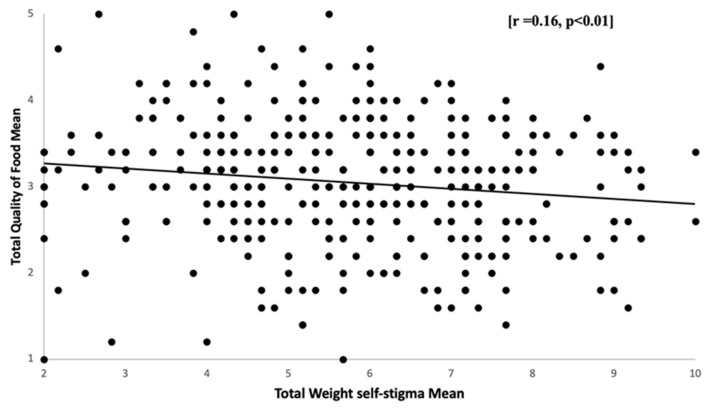
Relationship between total weight self-stigma mean scores and total quality of food mean scores: Scatter plots showing a correlation between students’ mean scores on quality of food and WSS. *n* = 385.

**Figure 7 healthcare-11-01754-f007:**
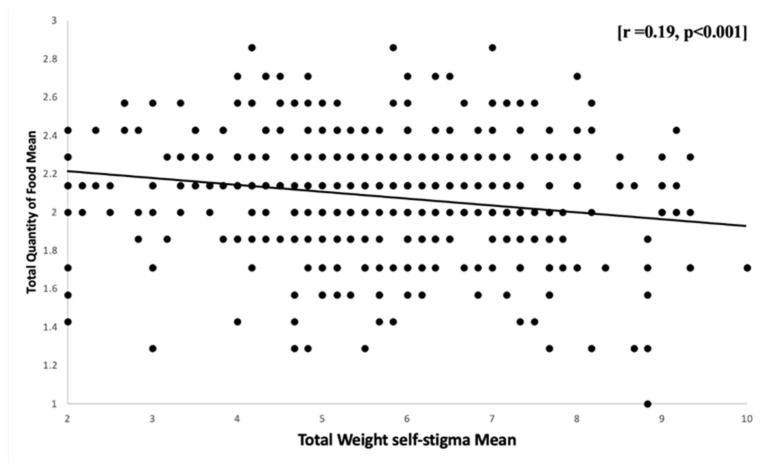
Relationship between total weight self-stigma mean scores and total quantity of food mean scores: Scatter plots showing a correlation between students’ mean scores on the quantity of food and WSS. *n* = 385.

**Table 1 healthcare-11-01754-t001:** Baseline characteristics of the sample.

	Characteristics	*n*	%
	Age in Years, (M ± SD)	6.74 ± 24	
Sex	Male	83	21.6
	Female	302	78.4
Educational Level	Postgraduate Education	104	27
	Bachelor’s degree	202	52.5
	High school or diploma or less	79	20.5
Nationality	Saudi	375	97.4
	Non-Saudi	10	2.6
Marital status	Single	325	84.4
	Married	55	14.3
	Divorced	5	1.3
Monthly income	More than 20,000 SAR	90	23.4
	SAR 15,000–20,000	81	21
	SAR 10,000–15,000	70	18.2
	SAR or less 10,000	144	37.4
Residency location	City	369	95.8
	Governorate	16	4.2
BMI Classification	Underweight (BMI < 18.5 kg/m^2^)	34	8.8
	Normal weight (18.5 ≤ BMI < 25.0 kg/m^2^)	170	44.2
	Overweight (25.0 ≤ BMI < 30.0 kg/m^2^)	92	23.9
	Obesity class I (30.0 ≤ BMI < 34.5 kg/m^2^)	47	12.2
	Obesity class II (35.0 ≤ BMI < 39.5 kg/m^2^)	21	5.5
	Obesity class III (BMI ≥ 40.0 kg/m^2^)	17	4.4

*n*: frequency; M: mean; SD: standard deviation; BMI: body mass index.

**Table 2 healthcare-11-01754-t002:** The weight self-stigma (WSS) questionnaire.

Items		Responses				
		Completely Agree	Agree	Neutral	Disagree	Completely Disagree
*n* (%)						
Self-devaluation						
1	I will always go back to being overweight	58 (15.1%)	81 (21%)	89 (23.1%)	97 (25.2%)	60 (15.6%)
2	I caused my weight problems	94 (24.4%)	135 (35.1%)	65 (16.9%)	55 (14.3%)	36 (9.4%)
3	I feel guilty because of my weight problems	76 (19.7%)	103 (26.8%)	69 (17.9%)	83 (21.6%)	54 (14%)
4	I became overweight because I am a weak person	66 (17.1%)	77 (20%)	84 (21.8%)	83 (21.6%)	75 (19.5%)
5	I would never have any problems with weight if I was stronger	113 (29.4%)	95 (24.7%)	86 (22.3%)	52 (13.5%)	39 (10.1%)
6	I do not have enough self-control to maintain a healthy weight	62 (16.1%)	93 (24.2%)	82 (21.3%)	82 (21.3%)	66 (17.1%)
Fear of enacted stigma						
7	I feel insecure about others’ opinions of me	49 (12.7%)	56 (14.5%)	89 (23.1%)	102 (26.5%)	89 (23.1%)
8	People discriminate against me because I have had weight problems	30 (7.8%)	25 (6.5%)	68 (17.7%)	119 (30.9%)	143 (37.1%)
9	It is difficult for people who have not had weight problems to relate to me	70 (18.2%)	75 (19.5%)	104 (27%)	62 (16.1%)	74 (19.2%)
10	Others will think I lack self-control because of my weight problems	65 (16.9%)	58 (15.1%)	106 (27.5%)	70 (18.2%)	86 (22.3%)
11	People think that I am to blame for my weight problems	61 (15.8%)	70 (18.2%)	95 (24.7%)	65 (16.9%)	94 (24.4%)
12	Others are ashamed to be around me because of my weight	14 (3.6%)	20 (5.2%)	69 (17.9%)	83 (21.6%)	199 (51.7%)

WSS: weight self-stigma.

**Table 3 healthcare-11-01754-t003:** Weight self-stigma (WSS) in the study population in terms of BMI, and sex.

Variables/Categories		Self-Devaluation			Fear of Enacted Stigma		
Adult Students		(M ± SD)	SE	*p*-Value	(M ± SD)	SE	*p*-Value
BMI classifications	Underweight	2.46 ± 0.90	0.15	^$^ 0.000	2.19 ± 0.91	0.15	^$^ 0.000
	Normal weight	2.80 ± 0.89	0.06		2.22 ± 0.83	0.06	
	Overweight	3.53 ± 0.88	0.09		2.78 ± 0.93	0.09	
	Obesity class I	3.81 ± 0.63	0.09		2.99 ± 0.86	0.12	
	Obesity class II	3.96 ± 0.84	0.18		3.43 ± 0.90	0.19	
	Obesity class III	3.66 ± 0.88	0.21		3.31 ± 0.87	0.21	
Sex	Male	3.41 ± 0.91	0.10	^#^ 0.132	2.76 ± 0.94	0.10	^#^ 0.725
	Female	3.11 ± 0.99	0.05		2.51 ± 0.95	0.05	
Educational Level	Postgraduate Education degree	3.14 ± 1.04	0.10	^$^ 0.728	2.55 ± 1.01	0.09	^$^ 0.882
	Bachelor’s degree	3.21 ± 1.00	0.07		2.55 ± 0.96	0.06	
	High school or diploma	3.12 ± 0.87	0.09		2.62 ± 0.88	0.09	

M: mean; SD: standard deviation; BMI: body mass index, SE: standard error, # indicates *t*-test has been used whereas $ indicates ANOVA test has been used.

**Table 4 healthcare-11-01754-t004:** Correlation between quality of life and weight self-stigma.

Quality of Life Domain	(M ± SD)	Weight Self-Stigma (r)
Physical domain	3.48 ± 0.73	−0.35 **
Psychological domain	3.20 ± 0.82	−0.41 **
Social domain	3.42 ± 0.75	−0.24 **
Environmental domain	3.33 ± 1.09	−0.23 **
Overall quality of life and general health	3.56 ± 0.97	−0.40 **

M: mean; SD: standard deviation. ** *p* < 0.01.

**Table 5 healthcare-11-01754-t005:** Frequency and percentage of dietary habits.

	*n* (%)				
	Excellent	Very Good	Good	Fair	Poor
1. How would you rate your overall habits of eating healthy foods, for example, eating a balanced meal of protein, fat, and carbohydrates?	17 (4.4)	50 (13)	106 (27.5)	81 (21)	131 (34)
	Daily	1–3 times/week	4–6 times/week	I do not eat home-cooked meals	
2. How often do you eat home-cooked meals per week?	143 (37.1)	94 (24.4)	130 (33.8)	18 (4.7)	
	Daily	1–3 times/week	4–6 times/week	I do not order from restaurants	
3. How often do you order from a restaurant, takeaway, or delivery per week?	29 (7.5)	265 (68.8)	43 (11.2)	48 (12.5)	

*n* = frequency.

## Data Availability

The raw data supporting the conclusions of this article will be made available by the authors, without undue reservation.

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
