# Peer review of "Effect of Weight Self-Stigma on Quality of Life and Dietary Habits among Adult Students in Riyadh, Saudi Arabia"

_healthcare, 2023, doi:10.3390/healthcare11121754_

Round 1
Reviewer 1 Report
The aim of the manuscript (ID: healthcare-2395549) was to examine the effect of weight self-stigma (WSS) on the quality of life (QOL) and dietary habits among Saudi students in the Riyadh region. It is necessary to enter the following corrections (major revision):
- Lines 2-3: Specify in the title of the manuscript who were the participants in this study: `adults`, `students`, or `adults students`;
- Throughout the entire manuscript state uniformly who the participants were in this manuscript;
- Lines 28-34: Use the appropriate references for the definitions of the stated items (WSS, etc), as follows:
- Corrigan PW, Larson JE, Rüsch N. Self-stigma and the "why try" effect: impact on life goals and evidence-based practices. World Psychiatry. 2009 Jun;8(2):75-81. doi: 10.1002/j.2051-5545.2009.tb00218.x. PMID: 19516923; PMCID: PMC2694098.
- Palmeira L, Cunha M, Pinto-Gouveia J. The weight of weight self-stigma in unhealthy eating behaviours: the mediator role of weight-related experiential avoidance. Eat Weight Disord. 2018 Dec;23(6):785-796. doi: 10.1007/s40519-018-0540-z. Epub 2018 Jul 17. PMID: 30019257.
- Lines 43-44: For the definition of the QOL by WHO, it is necessary to cite the appropriate reference;
- Lines 53-54: Define the abbreviation `PA`;
- Lines 62-71: Add the text that will describe the aimed `Study population`;
- Line 76: Add the definition and classification for Body Mass Index, with the citation of an appropriate reference;
- Lines 85-91: The cited reference (reference No 11) is not appropriate. Cite the references which show the psychometric characteristics of the Arabic version of the WHOQOL-BREF questionnaire, for example:
- Ohaeri JU, Awadalla AW. The reliability and validity of the short version of the WHO Quality of Life Instrument in an Arab general population. Ann Saudi Med. 2009 Mar-Apr;29(2):98-104. doi: 10.4103/0256-4947.51790. PMID: 19318760; PMCID: PMC2813624.
- Abdel-Khalek AM. Quality of life, subjective well-being, and religiosity in Muslim college students. Qual Life Res. 2010 Oct;19(8):1133-43. doi: 10.1007/s11136-010-9676-7. Epub 2010 Jun 29. PMID: 20585988.
- Lines 108-109: The paper also presents results for p<0.001. Correct this;
- Lines 252-258: Unnecessary repetition of the study aims. Delete.;
- Lines 329-336: Add more to this paragraph with the discussion whether the possible limitations of this study were sample size, lack of the assessment of some variables that were not included in this research (such as psychological distress, success during studies, employment, way of living, etc).
The quality of English language is appropriate.
Author Response
Amendment Letter
Manuscript ID healthcare-2395549
Title Effect of Weight Self-Stigma on Quality of Life and Dietary Habits Among Adults in Riyadh, Saudi Arabia
Track Changes option is activated
|
Comment |
Amendment |
|
Reviewer 1 |
|
|
The title was modified to: Effect of Weight Self-Stigma on Quality of Life and Dietary Habits Among Adult Students in Riyadh, Saudi Arabia
|
|
Throughout the entire manuscript state uniformly who the participants were in this manuscript |
Modified to students/ adult students |
|
The reference (Khodari, B.H., et al., The Relationship Between Weight Self-Stigma and Quality of Life Among Youth in the Jazan Region, Saudi Arabia. Cureus, 2021. 13(9): p. e18158.) was removed and the two stated references that define the WSS were added. |
|
The definition of QOL stated by the WHO QOL group was added “ an individual’s perception of their position in life in the context of the culture and value systems in which they live, and in relation to their goals, expectations, standards and concerns’’ Reference: Skevington SM, Sartorius N, Amir M, WHOQOL-Group1. Developing methods for assessing quality of life in different cultural settings: The history of the WHOQOL instruments. Social psychiatry and psychiatric epidemiology. 2004 Jan;39:1-8. |
|
Lines 53-54: Define the abbreviation `PA` |
Abbreviation is defined as physical activity |
|
Lines 62-71: Add the text that will describe the aimed `Study population` |
The study population targeted male and female university students aged 18 years and above attending a university in Riyadh |
|
Line 76: Add the definition and classification for Body Mass Index, with the citation of an appropriate reference |
Body mass index (BMI) was calculated as weight (kg) divided by height (m2) and recruited students were classified as underweight (BMI < 18.5 kg/m2), normal weight (18.5 ≤ BMI < 25.0 kg/m2), overweight (25.0 ≤ BMI < 30.0 kg/m2), obesity class I (30.0 ≤ BMI < 34.5 kg/m2), obesity class II (35.0 ≤ BMI < 39.5 kg/m2), obesity class III (BMI ≥ 40.0 kg/m2) conforming to WHO guidelines.
Reference: World Health Organization, The Global Health Observatory, Body Mass Index among Adults. [online] Available at: https://www.who.int/data/gho/data/themes/topics/indicator-groups/indicator-group-details/GHO/bmi-among-adults. 2023.
|
|
1. The reference (Baumann, M., Association between health-related quality of life and being an immigrant among adolescents, and the role of socioeconomic and health-related difficulties. International journal of environmental research and public health, 2014) is deleted and the two marked references were added.
|
|
Lines 108-109: The paper also presents results for p<0.001. Correct this |
P values of <0.001, < 0.01, or 0.05 were considered statistically significant.
|
|
Aims deleted |
|
The present study has certain limitations. Firstly, the study’s cross-sectional nature makes it difficult to better clarify the causal association between WSS and other factors and/or predictors. Secondly, the limited sample size may limit the ability to generalise results on the Saudi students and on other significant regions in Saudi Arabia. Considering that the survey was anonymously distributed to the target population, most respondents were females (78.4%), making it difficult to generalise our results, particularly on males, which require further investigation. Addressing the application of self-reported questionnaires, this study was expressed as one-stage research since no clinical assessments, interviews were applied. Moreover, no further assessment was established for some variables that could have a significant impact (such as psychological distress, success during studies, employment, way of living, etc). Despite that self-reported questionnaires are feasible and affordable tools, they might increase the probability of overestimation and bias. |

Reviewer 2 Report
Comments:
This study, "Effect of Weight Self-Stigma on Quality of Life and Dietary Habits Among Adults in Riyadh, Saudi Arabia", presents insightful information about the impact of Weight Self-Stigma (WSS) on the quality of life (QOL) and dietary habits among adult students in Riyadh, Saudi Arabia. By providing a detailed examination of the relationships between WSS, quality and quantity of food intake, and body mass index (BMI), the paper offers a nuanced understanding of the role of self-stigma in health behaviors and outcomes.
The authors’ use of various validated tools to measure WSS, QOL, dietary habits, and statistical methods lend credibility to their findings. Additionally, the large sample size of 385 students strengthens the reliability of the study results. The study findings echo those of previous research, suggesting a negative correlation between WSS and QOL and detrimental impacts on dietary habits. These findings underline the importance of treating WSS as a significant factor influencing health outcomes among overweight and obese individuals.
The paper effectively highlights the global nature of the WSS problem. The authors are also right in emphasizing the need for awareness and policy measures to mitigate WSS. The negative correlation between the quantity of food and the total score of WSS revealed in the study is particularly thought-provoking. The authors rightly suggest that this could be a result of misguided attempts to lose weight or a way to avoid negative judgments. The positive correlation observed between BMI and self-devaluation and fear of enacted stigma is also noteworthy. This strengthens the argument that individuals with higher BMIs are more susceptible to WSS.
A significant aspect of the study is its focus on the Saudi Arabian context, an area that may not have received sufficient attention in the literature. The data and discussion on the prevalence of obesity and overweight in the region add value to the study. This research offers a valuable contribution to the field, deepening our understanding of the effects of WSS on quality of life and dietary habits. These findings are critical for informing future interventions and policy-making in the context of obesity and weight management.
The authors have meticulously prepared this article. Overall the manuscript is well-written with updated references. The authors should be careful about the syntax and a few grammatical errors, which can be fixed with thorough editing.
Author Response
Amendment Letter
Manuscript ID healthcare-2395549
Title Effect of Weight Self-Stigma on Quality of Life and Dietary Habits Among Adults in Riyadh, Saudi Arabia
Track Changes option is activated
Thank you very much for your comments. They were very encouraging.
Alanoud Aladel

Reviewer 3 Report
Thank you very much for giving me the opportunity to review this manuscript. The idea of your article is interesting, my recommendations are the following:
Abstract
it would be recommended to include data disaggregated by sex. More in these cases where differentiating by sex is important, and let it be reflected in the summary if we are talking about male, female students or both
Introduction
It would be recommended to expand the introduction a little more with public social measures, impact of the topic, etc.
Methods
Study design and participants
it would be recommended to include the main and Sd of age.
it would be recommended to include data disaggregated by sex
Questionnaires
it would be recommended to include the reliability and validity of the questionnaires, including values ​​such as cronbach's afa
Results
it would be recommended to synthesize the results, especially with multiple scatterplots
References
It would be advisable to broaden references to include current works of interest, and also reflect the international
Author Response
Amendment Letter
Manuscript ID healthcare-2395549
Title Effect of Weight Self-Stigma on Quality of Life and Dietary Habits Among Adults in Riyadh, Saudi Arabia
Track Changes option is activated
|
Comment |
Amendment |
|
Reviewer 3 |
|
|
Abstract
it would be recommended to include data disaggregated by sex. More in these cases where differentiating by sex is important, and let it be reflected in the summary if we are talking about male, female students or both
|
Abstract: No significant difference was seen in study outcomes with respect to gender. Results: The majority of participants who volunteered to complete the questionnaire were female (302 (78.1%) of 385), with only 21.6% males recruited (n=83).
Prevalence of WSS: No significant differences were reported in relation to sex.
Self-devaluation: the mean self-devaluation did not differ significantly between males and females, though females significantly comprised the majority of the collected sample (78.4%). Fear of enacted stigma: No significant difference in fear of enacted stigma mean between males and females was determined according to independent sample t-test, though females significantly comprised the majority of the collected sample (78.4%). Quality of life and WSS: According to independent sample t-test, no significant difference in the QOL mean scores between males and females |
|
Introduction
It would be recommended to expand the introduction a little more with public social measures, impact of the topic, etc.
|
Society’s perception of weight stigma continues to exist because it is believed to be linked to personal self-control and responsibility, rather than into a combination of psychological, physiological, and genetic factor that plays a role in developing obesity [9,10]. This negative belief reinforces obese individuals as being lazy, unmotivated, lacking discipline, and more likely to suffer from social rejection and unfair treatment [11-14]. This defies the principles of social equality and diversity. Therefore, fighting weight stigma is crucial for human rights and for tackling obesity [15]. |
|
Methods Study design and participants
it would be recommended to include the main and Sd of age.
it would be recommended to include data disaggregated by sex
|
Adult students (n=385) included male (n=83, 21.6%) and female (n=302, 78.4%) university students (M age= 24, SD=6.74) from 10 different universities in Riyadh, Saudi Arabia, who have given their consent prior to enrolling in the study. |
|
Questionnaires
it would be recommended to include the reliability and validity of the questionnaires, including values ​​such as cronbach's afa
|
WSSQ: Therefore, it indicates that the Arabic WSSQ is a reliable tool for evaluating weight-related self-stigma in Arabic-speaking individuals (α=.982).
QOL: It is a self-administered questionnaire that reported significant reliability and validity indices among the Arab general population (α ≥0.7).
Dietary Habits Questionnaire: This Arabic dietary habits questionnaire was adopted from Alhusseini & Alqahtani’s [31] study and was approved for its content validity. The authors translated and adjusted the original version published in earlier studies to meet the Saudi population’s dietary requirements [32,33]. |
|
Results
it would be recommended to synthesize the results, especially with multiple scatterplots
|
The scatterplot (Figure 1) illustrates the linear relationship between the subject's mean scores on self-devaluation and BMI. There is a positive significant correlation between self-devaluation and BMI [r=0.47, p<0.001] indicating higher levels of self-devaluation with overweight and obesity.
Bar charts (Figure 2) showing self-devaluation mean and BMI of underweight subjects (2.4±0.9), normal weight (2.8±0.89), overweight (3.5±0.88), obese class I (3.8±0.63), obese class II (3.9±0.84), and obese class III (3.5±0.93). ). Compared to normal weight students, the mean scores for self-devaluation tend to be significantly higher in overweight (p<0.001)and obese (class I (p<0.001), class II (p<0.001), class III (p<0.01)) students.
The scatterplot (Figure 3) illustrates the linear and positive relationship between the adult students’ mean scores on the fear of enacted stigma and BMI [r =0.39, p<0.001] suggesting a significant and moderate increase on the fear of enacted stigma with higher BMI measures.
Bar charts (Figure 4) showing the fear of enacted stigma means and BMI of underweight subjects (2.1±0.91), normal weight (2.2±0.83), overweight (2.7±0.93), obesity class I (2.9±0.86), obesity class II (3.4±0.90), and obesity class III (3.2 ± 0.87). The fear of enacted stigma is confirmed to be significantly higher among overweight students (p<0.001), class I obese students (p<0.001), class II (p<0.001), and class III (p<0.001) compared to normal weight students.
The scatterplot (Figure 5a) demonstrates the linear and negative association between the adult students’ mean scores of QOL and BMI [r =0.11, p<0.05], indicating that QOL was significantly reduced with a higher BMI. Bar charts (Figure 5b) showing the quality of life and BMI of underweight subjects (16.1±3.86), normal weight (18±3.29), overweight (16.6±3.30), obese class I (16.1±3.78), obese class II (14.4±3.51), and obese class III (16.5±3.88). Scores of QOL tend to be significantly higher among normal weight students compared to overweight (p<0.05), class I obese (p<0.05), class II obese students (p<0.001). However, scores for class III obese students did not report a significant difference from normal weight students. According to the correlation between the QOL and WSS. The mean±SD of each subscale was described in Table 4. WSS correlated negatively with physical health domains [r= 0.35, p<0.001], psychological health domains [r= 0.41, p<0.001], social relationships domains [r= 0.24, p<0.001], environmental health domains [r= 0.23, p<0.001] and Overall QOL and General Health domains [r= 0.40, p<0.001]. The scatterplot (Figure 5c) illustrates the linear and negative association between adult students’ mean scores on the QOL and total WSS [r =0.40, p<0.001]. This means that QOL was significantly reduced in students who reported having WSS.
|
|
References
It would be advisable to broaden references to include current works of interest, and also reflect the international
|
References added (46 relevant references)
Discussion: Another study investigating WSS among 170 participants with obesity in Spain confirmed the positive association between WSS and BMI, depression, and anxiety [34]. A recent study in Hongkong concluded a significant and adverse association between WSS and poor health-related quality of life (r = −0.28 to −0.61) among overweight individuals [35].Moreover, current research focused on assessing the prevalence of WSS in adults aged 18 and older (n=3821). It was concluded that more than half of the sample (57%) has weight stigma and that the odds of WSS were the highest among overweight and obese adults [36]. |
|
General comments
|
|
|
Include at least 30 references, more extensive. Please check that all references are relevant to the contents of the
|
All points are addressed
References added
Track changes option is activated |

Round 2
Reviewer 1 Report
The authors addressed all my comments and made appropriate corrections in their manuscript. The revised version of this paper is clearer and more informative and provides important data for the topic it covers. Thanks to the authors.
The quality of English language is appropriate.